# A Different rTMS Protocol for a Different Type of Depression: 20.000 rTMS Pulses for the Treatment of Bipolar Depression Type II

**DOI:** 10.3390/jcm11185434

**Published:** 2022-09-16

**Authors:** Theodoros Koutsomitros, Kenneth T. van der Zee, Olympia Evagorou, Teresa Schuhmann, Antonis C. Zamar, Alexander T. Sack

**Affiliations:** 1Department of Cognitive Neuroscience, Faculty of Psychology and Neuroscience, Maastricht University, 6211 LK Maastricht, The Netherlands; 2Greek rTMS Clinic, Medical Psychotherapeutic Centre (ΙΨΚ), 546 24 Thessaloniki, Greece; 3Cyprus rTMS Clinic, 3082 Limassol, Cyprus; 4Brain Imaging Centre (MBIC), Maastricht University, 6229 EV Maastricht, The Netherlands; 5The London Psychiatry Centre, London W1G 7HG, UK; 6School for Mental Health and Neuroscience (MHeNs), Brain and Nerve Centre, Maastricht University Medical Centre+ (MUMC+), 6229 ER Maastricht, The Netherlands

**Keywords:** repetitive transcranial magnetic stimulation, bipolar disorder type II, bipolar depression, relapse, neuromodulation

## Abstract

In this open-label naturalistic study, we assess the feasibility, tolerability, and effectiveness of a repetitive transcranial magnetic stimulation protocol with a reduced total pulse number for treating patients suffering from bipolar disorder type II. All patients received one rTMS treatment session of 1000 pulses for 20 consecutive working days, accumulating to 20.000 rTMS pulses applied over 4 weeks. We measured the patients’ symptoms before the start, halfway through, directly after, and one month after treatment. We quantified the depression symptoms using both the Beck depression inventory scale and the symptom checklist-90 depression subscale. Patients showed a significant reduction in depression symptoms directly after treatment and an even further reduction one month after treatment. The remission rates were at 26% halfway through treatment (after the 10th session), 61% directly after treatment (after the 20th session), and increased to 78% at the 1-month follow-up. Importantly, the protocol proved to be feasible and highly tolerable in this patient population, with no adverse effects being reported. Considering these positive results, further research should focus on replicating these findings in larger clinical samples with control groups and longer follow-up periods, while potentially adding maintenance sessions to optimize the treatment effect and stability for bipolar disorder type II patients.

## 1. Introduction

Bipolar disorder (BpD) is a severe, recurrent, and chronic psychiatric disorder with an estimated population prevalence of 3% [1]. BpD is characterized by mood swings that include emotional highs (mania or hypomania) and lows (depression). There are also different subtypes of BpD.

For the diagnosis of bipolar type II (DSM code: 296.89, ICD10 code: F31.8), patients need to fulfill the diagnostic criteria of at least one current or past major depressive episode as well as the criteria of at least one current or past hypomanic episode while never having been diagnosed with a manic episode.

BpD greatly impairs the quality of life, social relationships, and occupational performance [2,3]. It is associated with cognitive dysfunction, including deficits in verbal learning and expression, working memory, attention, and psychomotor speed [4]. The loss of disability-adjusted life years in BpD ranks it among the highest out of all the mental and neurological disorders worldwide [5]. Given this tremendous burden of the disease, it is important to have effective treatment options for patients suffering from bipolar depression.

However, the treatment of BpD can be challenging due to the high risk of comorbid diseases, the high incidence of residual symptoms, the heavy burden of complex polypharmacy [1,6], and the delicate balance and switches between (hypo)manic and depressive episodes. Psychosocial (cognitive behavioral therapy, functional remediation, psychoeducation) and pharmacological treatments (antipsychotics, anticonvulsants, lithium) are the most common forms of therapy. There are several US Food and Drug Administration (FDA)-approved medications with level 1 efficacy for the management of acute mania and mixed episodes in BpD. However, the current FDA-approved evidence base for the treatment of bipolar depression is only limited to five drugs (olanzapine and fluoxetine combination, quetiapine, lurasidone, cariprazine, and lumateperone), and only one of these drugs (quetiapine) is licensed by the European Medicines Agency (EMA) in Europe for the treatment of bipolar depression [7,8]. Lurasidone and cariprazine have been available in many European countries for only a few years, and lumateperone was only approved by the FDA in December 2021. Practitioners may favor the use of antidepressants in patients with bipolar II disorder when the response to mood stabilizers is inadequate or in case of suicidal risks [9], although antidepressant use was found to be no different from placebo in the STP-BD study. Moreover, their use is associated with a significant risk of mood destabilization [10]. Psychostimulants, followed by modafinil/armodafinil and thyroid hormone, are also sometimes considered for bipolar depression treatment. The thyroid hormone improves cognitive function and is not associated with any side effects [11,12]. However, the other drugs do not significantly improve cognitive functioning and carry the risk of serious drug-induced side effects [9,13]. Therefore, more effective and safer interventions are needed in managing BpD.

Another problem with the use of medications, especially antidepressants impacting nerve cells and receptor sites over extended periods of time, is that it is difficult, if not impossible, to target only one pole of the bipolar disorder. Therefore, for all these various reasons, a non-pharmacological treatment alternative for BpD is clearly needed.

As one of the most promising upcoming alternatives to traditional drugs and/or psychotherapy, transcranial magnetic stimulation (TMS) has been at the center of interest in research and therapy [14]. Following several randomized clinical multi-center trials regarding its clinical efficacy, repetitive TMS (rTMS) has been acknowledged and approved by the FDA as a therapy for treatment-resistant depression (TRD) in 2008 [15]. Today, TMS is an approved treatment for depression in many countries and is recommended for this purpose by the American Psychiatric Association [16], the Canadian Network for Mood & Anxiety Disorders [17], the UK Maudsley Guidelines [18], and the World Federation of Societies of Biological Psychiatry [19]. rTMS is a focally applied noninvasive neuromodulation technique that can induce protocol-specific neuroplastic changes in the stimulated brain region. Furthermore, depending on the frequency with which the electromagnetic stimulation is applied, these neuroplastic changes can cause a longer-lasting increase (stimulate) or decrease (inhibit) in cortical excitability in the targeted brain area. It is by now established that the so-called high-frequency rTMS protocols generally cause an increase in neural activity and functional connectivity within the stimulated brain networks, whereas the so-called low-frequency rTMS protocols will cause the opposite effect, namely, reduced neural activity in these same networks [20,21]. Importantly, these effects require repeated stimulation with rTMS over several sessions, thereby inducing these neuroplastic changes that outlast the duration of the treatment itself due to changes in the synaptic transmission efficacy similar to LTP and LTD-like processes [22,23].

Based on this mechanism, clinical rTMS applications have mainly focused on the unipolar major depression disorder (MDD), initially testing the efficacy in treatment-resistant patients. The rationale for this focus on unipolar MDD was fueled by an assumption regarding the underlying pathophysiology in MDD in which depression was linked to the dorsolateral prefrontal cortex (DLPFC), its connections to the limbic system, and an (im)balance in baseline activity between the left and right DLPFC. In this assumption, the left DLPFC shows hypo-activity and would thus require an rTMS protocol aiming at increasing its neural activity levels. Accordingly, the primary TMS approach to treat depression today—which is also the one that received Food and Drug Administration (FDA) approval in the United States in 2008, as well as health insurance coverage in an increasing number of countries worldwide—is daily high-frequency repetitive TMS (HF-rTMS) applied to the left dorsolateral prefrontal cortex for several weeks [24,25,26].

However, while the feasibility, tolerability, and efficacy of rTMS for treating unipolar MDD has been established, the question of whether or not rTMS is also capable of effectively treating BpD, and if so, based on which exact protocol, is still open. Recent studies using rTMS in BpD have investigated various stimulation parameters, including high-frequency (>5 Hz) rTMS over the left DLPFC, low-frequency (1 Hz) rTMS over the right DLPFC, and sequential bilateral stimulation over the DLPFC, with treatment course durations varying between two and four weeks. A meta-analysis assessing 19 different randomized clinical trials in patients with bipolar disorder (*N* = 181) showed efficacy for left-sided high-frequency and right-sided low-frequency stimulation but not for sequential bilateral stimulation [27]. A most recent systematic review and meta-analysis regarding the efficacy of rTMS for bipolar disorder (*N* = 274) showed that the response rates were higher in rTMS compared to sham treatment (odds ratio OR= 2.72, 95%CI: 1.44–5.14). However, when the stimulation protocols were analyzed separately, a statistically significant clinical response was only observed for high-frequency rTMS over the left DLPFC (OR = 2.57, 95%CI: 1.17–5.66) [28].

In conclusion, it seems that high-frequency rTMS over the left DLPFC, and thus, the same protocol used for treating unipolar depression, seems to also be a promising treatment protocol for BpD. This is not surprising, considering the fact that although the presence of manic and hypomanic episodes differentiates and defines bipolar disorder as compared to unipolar depression, the syndromal and subsyndromal depressive symptoms are the ones causing the main burden of this illness and a substantial proportion of disability [29]. Patients are reported to spend as much as half of their lives with mood symptoms, with depressive symptoms accounting for 70% to 82% of the symptomatic periods [30].

Still, the main concern in using a high-frequency rTMS protocol over the left DLPFC in BpD is the potential risk of triggering or inducing a hypomanic, mixed, or even manic episode—in other words, causing a switch of polarity between episodes in these patients. However, it is unclear how high this risk really is in a naturalistic setting and whether this theoretical risk can be further reduced by also reducing the total number of pulses classically applied in an rTMS depression treatment protocol. After all, it seems reasonable to assume that rTMS may also offer a promising new treatment strategy for BpD, especially because the rTMS therapy aims to target different neural mechanisms compared to existing treatments by providing the opportunity of flexibly applying briefer, episode-based interventions for BpD without the risk of manic and hypomanic switches or mood destabilization. To address these questions, we used a specific rTMS depression protocol for treating BpD-II patients in which we reduced the number of pulses per session from the standard 3000 to 1000 pulses and also reduced the number of sessions from the standard 30 to 20, resulting in a total number of 20.000 pulses as compared to 90.000 pulses in the classical rTMS MDD treatment protocol [31]. By using this different rTMS protocol for a different type of depression, we aimed to investigate whether the application of this protocol in BpD is feasible, tolerable, and clinically effective.

Our rationale for testing this specific rTMS protocol in the current patient population was based on the described risk of inducing a hypomanic switch in bipolar patients when stimulating intensively with high-frequency rTMS. Although rare, such rTMS-triggered inductions of hypomania have been reported in the literature [32], and this population may be of particular risk in this regard. At the same time, recent findings suggest that the clinical efficacy of high-frequency rTMS sessions can also be achieved with a reduced number of pulses [33,34,35,36,37]. The combination of these two considerations formed the basis for our rationale to test the reduced total rTMS pulse application when treating bipolar patients with high-frequency rTMS described here.

## 2. Materials and Methods

### 2.1. Study Sample

The current study was an open-label naturalistic study. Patients with a BpD-II diagnosis were recruited at the Medical Psychotherapeutic Centre in Thessaloniki, Greece, between 2017 and 2019. In total, 23 individuals were screened for eligibility and were included in the study. All patients gave their written informed consent to participate and to have their data used for scientific research, prior to participation.

None of the patients was drug naïve, and they were all on a mood stabilizer, mostly second-generation antipsychotics. Most of the patients were either on Quetiapine (9 patients) or Olanzapine (8 patients) monotherapy without any other mood stabilizer. Five patients were on Lithium, and six patients were also using an antidepressant: five were on SSRIs (Fluoxetine or Escitalopram), and one was on SNRI (Venlafaxine).

The inclusion criteria were: (1) meeting the DSM-5 criteria for bipolar II disorder (296.89) as their primary diagnosis using the structured clinical interview for DSM-5 disorders—clinician version (SCID-5-CV), as well as ICD10 criteria for other bipolar affective disorders (F31.8); (2) aged between 20 and 65 years; (3) fulfilling the ICD10 criteria for a severe depressive episode without psychotic symptoms. The exclusion criteria were: (1) primary diagnosis other than bipolar depression; (2) standard rTMS contraindications (history of epilepsy, ferromagnetic head implants, history of neurosurgical operations, or a pacemaker implant); (3) use of medication known to substantially lower the threshold of seizures (e.g., clozapine); (4) co-initiation of any medication, since the literature is too limited in order to inform synergistic or additive effects.

### 2.2. Study Sample Subgroups

Since this was an open-label naturalistic study, the patient sample was rather heterogenous, as indicated in Table 1 below. Out of the total 23 patients (12 women), 13 patients had, in addition to an ICD10 diagnosis of BpD, comorbidity of personality disorder according to ICD10, whereas the other 10 did not have any personality disorders on Axis-II. Eleven patients were using benzodiazepines (alprazolam, bromazepam, lorazepam, diazepam) during their rTMS therapy, whereas twelve patients did not use any benzodiazepine during their rTMS treatments. None of the eleven patients on benzodiazepines used more than 4 mg equivalent of lorazepam per day.

### 2.3. Procedure of rTMS

Patients who were included in our open-label naturalistic study had been treated at the Medical Psychotherapeutic Centre for at least 5 years. Before we included them as potential candidates for rTMS therapy, they all completed the BDI and SCL-90-R questionnaires to objectively assess the severity of their current depressive episode. In order to apply rTMS, we used a TMS stimulator connected to a figure of eight coil (MagVenture R20 and R30; MC-B70). The TMS coil was positioned over F3 of the international 10–20 system for electroencephalography. To this end, after defining F3 according to the 10–20 system, this point was marked on a cap placed on the head of the participant relative to the anatomical landmarks of the patient’s skull (nasion-inion) to reliably target F3 during each following treatment session. Next, the resting motor threshold (MT) was determined by applying single pulses of TMS in a steadily decreasing intensity over the right motor cortex. When 5 out of 10 stimuli resulted in a muscular contraction in the left FDA muscles, this stimulation intensity was taken as MT. The actual rTMS treatment was given at an intensity of 120% of the resting MT. During each rTMS treatment session, a 10 Hz protocol was applied with 50 pulses per train in 20 trains, with an inter-train interval of 11 s at 120% MT, resulting in 1000 pulses per session (5 min total treatment time) and 20,000 pulses over 20 consecutive working days with daily rTMS sessions. This reduced number of pulses per session and pulses in total as compared to the standard rTMS depression protocol was used to limit the risk of inducing (hypo)manic episodes in BpD.

### 2.4. BDI and SCL-90-R Questionnaires

In order to assess the clinical efficacy, the Beck depression inventory (BDI) and the symptom checklist-90 revised (SCL-90-R) were given to the patient sample. The BDI consists of a self-report scale and includes 21 items. The score results from the sum of the scores for each question. The maximum score is 63. Scores of 29 and above indicate severe depressive disorder [38]. Internal consistency has been confirmed by numerous studies [39,40,41].

SCL-90-R, developed by Leonard R. Derogatis, is a widely used questionnaire, covering 90 symptoms, which are divided into nine dimensions: somatization, obsessive compulsive disorder, interpersonal sensitivity, depression, anxiety, hostility, phobic anxiety, paranoid ideation, and psychoticism [42]. It is a self-report clinical rating scale, which is scored on a 5-point scale from 0 to 4, and the scores for each dimension are means of the scores for all items of the dimension. It is considered as a tool with good reliability although controversial validity [43].

### 2.5. Statistical Procedure

Linear mixed model analyses were used to analyze the treatment effect, as assessed by the BDI scale and SCL subscale. These were favored over repeated-measures analysis of covariance, since the latter filters out incomplete case files. In the case of our study, only the subgroup of patients who completed the treatment and were checked at follow-up would be included in the analysis (*n* = 9) compared to all patients (*N* = 23) when making use of the linear mixed models. Thus, two linear mixed models were built, which, respectively, used BDI or SCL as their dependent variable; all measurements before, during, and after treatment as the repeated factor ‘session’; and the factors of age, personality disorders, benzodiazepines, incidental bidaily treatment, and one of the respective baseline scores as the covariates. Post hoc Bonferroni corrected pairwise comparisons were performed to analyze the treatment effect and its stability at follow-up.

## 3. Results

### 3.1. Tolerability

No side effects were reported, except for one female patient who stopped treatment at the fourteenth rTMS session due to a self-reported increase in irritability. Only 3 more out of 23 patients dropped out in the middle of the therapy (eleventh and twelfth rTMS session), mainly because they were not able to travel to the Greek rTMS clinic daily or afford to stay in Thessaloniki for the duration of the treatment, resulting in a dropout rate of 17%. Overall, our data provide proof that the HF-rTMS treatment can be used in BpD-II patient populations without eliciting adverse effects in most patients.

### 3.2. Clinical Efficacy

Both the BDI scale and the SCL-90-R depression subscale showed that the depression symptoms decreased significantly as a consequence of the rTMS treatment. To be more specific, at the start of the study, the patients had an average BDI score of 26.26 and an average SCL subscale score of 28.09. After ten treatment sessions, and thus halfway through the therapy, the BDI scores were already significantly reduced to 19.39, while the SCL scores were insignificantly reduced to 22.65. Directly after treatment (after 20 sessions), the scores were significantly reduced to 14.83 and 18.50, respectively. The treatment effects were shown to further increase one month post-therapy, with patients scoring an average of 7.67 and 11.89, respectively. Directly after treatment, 61% of our BpD patients were in full remission, based on a cut-off rate of 13 ≤ BDI score [44]. Remarkably, 26% of patients had already entered remission after 10 treatment sessions. One month after the end of treatment, the remission rate even increased to 78%, offering another indication that the treatment effect remained stable and even continued to improve over time.

### 3.3. Beck Depression Inventory

In addition to analyzing the isolated effect of treatment, we included variables as covariates, since they could have a confounding influence on the treatment effect. The first possible confounder integrated into the model as a covariate was the ‘pre-test score’ of each patient, since the severity of the depression symptoms at the start of treatment could have influenced the response to treatment. Another factor that was used as a covariate was ‘age’, since increasing age has been observed to cause a decrease in rTMS responsiveness [45]. A third possible covariate that was integrated into the model was the use of ‘any benzodiazepines’, since this can negatively influence the effectiveness of the rTMS treatment. One other factor that can influence the depression recovery rate of patients that is commonly found in patient groups with BpD-II is the comorbidity of other personality disorders [46]. The last factor that was included as a covariate that might have impacted the treatment effect was that some patients could sometimes only attend sessions every second day, which led to them receiving an equal amount of treatment over a slightly longer period, from here on referred to as ‘bidaily treatment’.

A linear mixed model was used to determine the significance of the effect of treatment and whether any of the covariates had a confounding effect on the dependent variable. The ‘patient’ factor was used to identify subjects; the four timepoints were included as clusters by using the session as the repeated factor. To compensate for the influence of pre-test scores on the treatment response, these were used as a covariate in addition to all the previously mentioned covariates and their interaction with the ‘session’ factor. All covariates were included as fixed effects in the first version of the model. The identity covariance matrix was used since the model became unidentifiable when using any other covariance matrices. The fixed effects of the resulting model are shown in Table 2 and the BDI score change over the course of treatment are illustrated in Figure 1. In order to determine the treatment effect when comparing individual sessions, post hoc pairwise comparisons were conducted, and the results are shown in Table 3.

When analyzing the effects of the covariates, the model shows that only the pre-test scores had a significant main effect and a significant interaction with the different sessions. Contrary to the evidence from previous studies, neither age, benzodiazepine use, nor personality disorders comorbidity had a significant influence on the treatment response. In addition, bidaily treatment did not seem to affect the treatment response either, but this might have been caused by the fact that these patients still received 20 sessions. The pairwise comparisons showed that the treatment effect was already significant after the 10th treatment session; it was still significant after the 20th treatment session, and it stabilized and increased when comparing the 20th treatment session to the results at 1-month follow-up.

### 3.4. Symptom Checklist-90-Revised Depression Subscale

In addition to testing the effect the rTMS treatment had on the BDI score of patients, the effectiveness of the treatment was also measured by analyzing the change in the SCL-90-R depression subscore. First, it was analyzed whether this subscale found a similar reduction in depression symptoms between the pre-test and the end of treatment. The way to accomplish this was to employ another linear mixed model, partly to, again, use the whole dataset instead of just the cases with a follow-up measurement (*n* = 9) and to analyze whether any of the possible confounders had a significant influence on the SCL score. Each of these factors was analyzed as a fixed effect with the addition of every possible confounder’s interaction with the session factor also being tested as a fixed effect. Only two models could be identified—one with the compound symmetry and one with the identity covariance matrix. Using −2 log-likelihood ratio tests, it was determined that the identity covariance structure (*df* = 27, REML = 424.118) fitted the data equally well as the compound symmetry (*df* = 26, REML = 424.118) structure while being more parsimonious.

The dependent variable was the SCL score; the repeated random factor consisted of the four levels of session, and the patient was used as a random interval. The covariates of pre-test score, age, benzodiazepine use, personality disorders, and bidaily treatment, together with the factor of session, were implemented as fixed factors. Thus, the final model used an identity covariance structure with all fixed factors and their interactions. The factors and their effects on the outcome measure of SCL are shown in Table 4 and the change in SCL score over the course of treatment is illustrated in Figure 2. Similar to the results from the BDI model, this model found that only the influence of the pre-test and its interaction with session had a significant influence on the outcome measure throughout treatment and during follow-up. What can be derived from this is that the baseline score of a patient determined their response to treatment according to both outcome measures. None of the other factors had a significant influence on the treatment response or the treatment response over the course of treatment according to both outcome measures.

Post hoc pairwise comparisons were conducted to determine the exact treatment effect betwen the sessions. The results of this test are shown in Table 5. Only some of the results are similar to those found using the BDI outcome measure. The difference between the pre-test and the 20th session and follow-up measurements is significant, indicating that the effect the treatment had on the SCL score was significant. In addition, the effect again seems to have stabilized and further increased when comparing the results after the 20th session and follow-up. The difference between the pre-test and the 10th session was, unlike the result of the BDI scale, insignificant; it indicated that 10 sessions of treatment are insufficient to significantly impact the SCL scores.

## 4. Discussion

We showed that using a specific rTMS protocol with a reduced number of pulses per session is tolerable and clinically effective in treating bipolar disorder patients. Our protocol of 1000 pulses applied in trains of 10 Hz only lasted for 5 min per session, with daily treatment sessions over 20 working days (4 weeks). As a result, our 23 patients with bipolar depression type II significantly improved, with an overall remission rate of 61% after 20 treatment sessions and a remission rate of even 78% at one-month follow-up. Importantly, no patient experienced any severe adverse effects, except one patient who dropped out due to increased irritability. We did not induce a hypomanic episode in any of our patients. These results are thus indicative of the fact that rTMS is also safe and highly tolerable in bipolar depression and leads to similar and even higher response and remission rates as described in the approved and recognized unipolar depression rTMS therapies. Therefore, our rTMS study clearly supports the notion that rTMS is an effective alternative and complementary to medication in the management of bipolar depression. 

As a side finding, it is interesting to note that according to the BDI score, the reduction was already significant after the 10th session, in addition to the significant difference found after 20 sessions and the stabilization of that effect after one month for both the BDI and SCL scales. One other factor that was found to influence both outcomes and their change over the course of treatment was the patients’ scores from the pre-test, which indicates that the symptoms upon treatment admission significantly influence the treatment response. Another factor to consider is that of patients receiving occasional bidaily treatment, i.e., not daily but only every other day rTMS, despite receiving the same total number of sessions over time (20 sessions—5 per week, on average). This was analyzed, since it was shown that it is better to offer daily or even slightly accelerated rTMS protocols instead of temporally wider-spaced treatments. In our patient population, no significant effect or interaction effect of the sessions was found, which means that occasionally moving a treatment session by one day does not affect the outcome of the treatment [24].

One of the problems in developing effective and safe treatments for bipolar depression is that the population is quite heterogeneous, with subgroups that possibly require different pharmacological interventions [9,47]. Regarding the heterogeneity of a naturalistic patient sample, we identified a priori several possible confounders, which we also tested statistically as covariates in our model. These confounders included age, comorbid personality disorders, and the use of benzodiazepine. In clinical populations suffering from BpD-II with long-term depression, the medication known as benzodiazepines is commonly used, as was also evident in our sample (Table 1). The use of benzodiazepines is known to decrease the reliability of the rTMS treatment, since this medication is known to decrease the clinical efficacy of rTMS treatments similar to the one applied here [48].

However, in our study, benzodiazepine did not affect the outcome of rTMS. This may be explained by medication dosage, since the total dosage used was less than 4 mg lorazepam equivalent, within the usual British National Formulary (BNF) prescription guidelines, and this could be the possible cause for it having an insignificant interaction with the treatment effect, as previous studies have shown. Likewise, the non-significant influence of age on treatment efficacy in our study was most likely caused by the relatively high mean age and low variation in age within this population (M = 56.390, SD = 11.264). It is important to underline this, since age is known to affect rTMS response when looking at groups with a wider age range. In line with the previous studies [49,50,51], personality disorders did not have a significant effect at all. 

Despite the reported overall very positive and highly promising findings of this specific rTMS protocol for treating bipolar disorder, including high efficacy (61% remission directly after treatment and 78% at 1-month follow-up) and tolerability (no adverse effects), it is important to point out that the results of this open-label naturalistic study should be interpreted with caution due to the intrinsic limitations of such designs. A naturalistic open-label study means that, here, we showed the data of actual patients (heterogeneous in terms of comorbidity and medication) with all the associated advantages compared to highly controlled and selective clinical trials. However, unlike in randomized clinical trials, our study did not include a placebo control treatment arm and, as such, the treatment effects are unavoidably confounded/mixed with placebo effects. Considering that the placebo effects in rTMS are prominent and considerably high, the response and remission rates reported here are certainly higher than those achieved in placebo-controlled trials. In addition, all assessments for evaluating a patient’s depressive symptoms and responses to rTMS were self-reporting tools. Evaluations based on an objective clinical assessment performed by a psychiatrist, such as the Hamilton depression rating scales, were omitted. However, despite these limitations and reasons for caution, the remission rates reported here were much higher than the commonly reported placebo effects in randomized placebo-controlled trials using rTMS in unipolar depression, and therefore, it is highly unlikely that these results were solely caused by the placebo. In this context, it is also important to note that the patients included here were treatment resistant, and thus showed no placebo effects of any therapy before, and that the clinical efficacy increased even further after the end of the acute rTMS treatment, accumulating in a remission rate of 78% one month after treatment discontinuation. All of this makes us confident that these effects are not due to the placebo alone, although a proper placebo condition should still be included in a future randomized control trial replicating our open-label findings.

## 5. Conclusions

This open-label naturalistic study showed the potential efficacy of high-frequency rTMS with a reduced number of pulses applied over the left DLPFC in patients with bipolar depression type II. Given that the overall treatment effect was highly significant, with a remission rate of 61% after 20 sessions and 78% at one-month follow-up, and given that no side effects, especially no induced hypomanic episodes, were induced, these results are highly encouraging and motivate replication studies using a randomized control trial design.

## Figures and Tables

**Figure 1 jcm-11-05434-f001:**
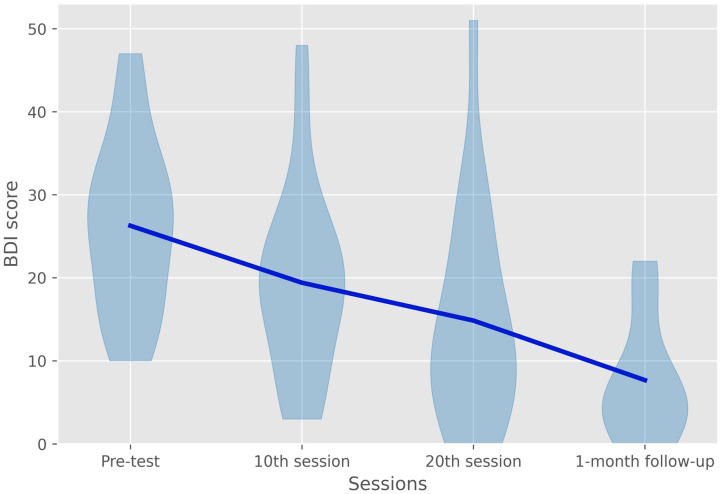
Beck depression inventory score before, during, and after treatment and during follow-up. The overall effect of treatment was significant (*MD* = −10.982, *df* = 34.776, *p* = < 0.001). Additionally, the treatment effect remained significant when comparing the 20th session to the 1-month follow-up measurement (*MD* = −7.205, *df* = 38.789, *p* = 0.479). Interestingly, BDI scores were shown to already be reduced significantly after ten sessions of treatment (*MD* = −7.176, *df* = 33.028, *p* = 0.009).

**Figure 2 jcm-11-05434-f002:**
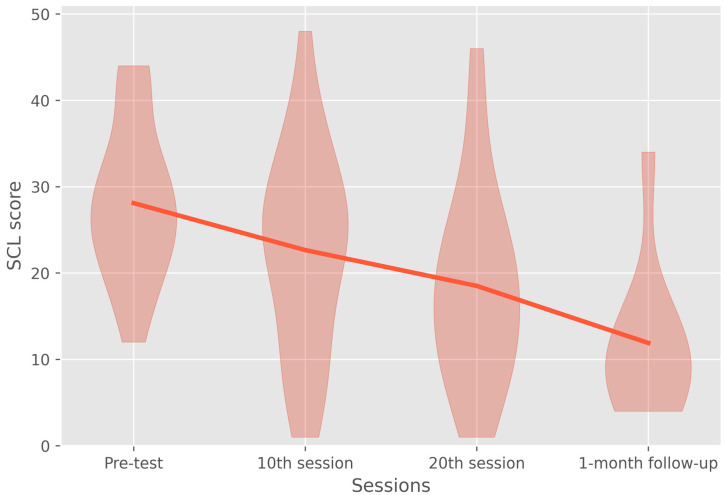
Beck depression inventory score before, during, and after treatment and during follow-up. The overall effect of treatment was significant (*MD* = −11.167, *df* = 17, *p* = 0.001). Additionally, the treatment effect remained significant when comparing the pre-test to the 1-month follow-up measurement (*b* = 4.949, *t*(11.749) = 4.979, *p* = 0.000). Interestingly, BDI scores were shown to already be reduced significantly after ten sessions of treatment (*MD* = −6.870, *df* = 22, *p* = 0.001).

**Table 1 jcm-11-05434-t001:** Sample demographics.

Any Personality Disorder		Any Benzodiazepines		Total
		No	Yes	
No		7	3	10
	Females	4	0	4
	Males	3	3	6
Yes		5	8	13
	Females	1	7	8
	Males	4	1	5
Total		12	11	23
	Females	5	7	12
	Males	7	4	11

**Table 2 jcm-11-05434-t002:** F-statistics for the parameters in the linear mixed model of BDI.

Parameter	Numerator *df*	Denominator *df*	*F*-Statistic	*p*-Value
Session	3	35.99	0.717	0.549
Any personality disorder	1	31.002	0.002	0.964
Benzodiazepine use	1	31.918	0.346	0.561
Bidaily treatment	1	19.059	0.153	0.7
Age	1	32.467	0.478	0.494
Pre-test	1	22.468	9.566	0.005
Session * Any personality disorder	3	35.564	0.08	0.97
Session * Benzodiazepine use	3	35.632	0.58	0.632
Session * Bidaily treatment	3	36	0.576	0.634
Session * Age	3	36.52	0.576	0.634
Session * Pre-test	3	35.938	6.187	0.002

* The asterisk refers to the interaction between the two parameters.

**Table 3 jcm-11-05434-t003:** Pairwise comparisons between different levels of session for the linear mixed model of BDI.

Session		Mean Difference	*df*	*p*-Value
Pre-test	10th session	7.176	33.028	0.009
	20th session	10.982	34.776	<0.001
	Follow-up	18.187	38.873	<0.001
10th session	Pre-test	−7.176	33.028	0.009
	20th session	3.806	34.776	0.614
	Follow-up	11.011	38.873	0.050
20th session	Pre-test	−10.982	34.776	<0.001
	10th session	−3.806	34.776	0.614
	Follow-up	7.205	38.789	0.497
Follow-up	Pre-test	−18.187	38.873	<0.001
	10th session	−11.011	38.873	0.050
	20th session	−7.205	38.789	0.497

**Table 4 jcm-11-05434-t004:** F-statistics for the parameters in the linear mixed model of SCL.

Parameter	Numerator *df*	Denominator *df*	*F*-Statistic	*p*-Value
Session	3	35.975	1.723	0.180
Any personality disorder	1	33.678	1.978	0.169
Benzodiazepine use	1	33.317	0.807	0.375
Bidaily treatment	1	18.855	0.000	0.995
Age	1	32.737	0.184	0.671
Pre-test	1	26.764	6.476	0.017
Session * Any personality disorder	3	36.199	0.963	0.420
Session * Benzodiazepine use	3	36.324	0.813	0.495
Session * Bidaily treatment	3	35.890	0.284	0.837
Session * Age	3	37.194	0.096	0.962
Session * Pre-test	3	36.695	5.309	0.004

* The asterisk refers to the interaction between the two parameters.

**Table 5 jcm-11-05434-t005:** Pairwise comparisons between different levels of session for the linear mixed model of SCL.

Session		Mean Difference	*df*	*p*-Value
Pre-test	10th session	5.322	33.441	0.099
	20th session	9.814	35.353	<0.001
	Follow-up	15.607	39.647	0.002
10th session	Pre-test	−5.322	33.441	0.099
	20th session	4.492	35.353	0.355
	Follow-up	10.285	39.647	0.076
20th session	Pre-test	−9.814	35.353	<0.001
	10th session	−4.492	35.353	0.355
	Follow-up	5.793	39.776	0.959
Follow-up	Pre-test	−15.607	39.647	0.002
	10th session	−10.285	39.647	0.076
	20th session	−5.793	39.776	0.959

## Data Availability

Data available on request due to restrictions. The data presented in this study are available on request from the corresponding author.

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
