# Peer review of "A Different rTMS Protocol for a Different Type of Depression: 20.000 rTMS Pulses for the Treatment of Bipolar Depression Type II"

_jcm, 2022, doi:10.3390/jcm11185434_

Round 1
Reviewer 1 Report
The current study investigated the modified rTMS protocol with reduction of 20000 pulses in the treatment of bipolar II depression. The results showed that the modified rTMS protocol may improve the severity of depressive symptoms and have the remission rate of 78% at one month after treatment. However, the study was an open-label study. Because rTMS had prominent placebo effect, the results should be interpretated with caution. The current study did not have clear rationale to reduce rTMS pulses to 2000 pulses in treating bipolar II depression. The medication effects were not clear, either. In addition, there were several points need to be addressed.
1. Lack of reference in Page 2, line 55-56.
2. The citations of references were wrong in the manuscript.
3. The rationale to reduce the total rTMS treatment pulses to 20000 pulses is not clear. As the reference is wrong, the readers can not understand the rationale or finding data supporting the reduction protocol.
4. Were the participants drug naïve and free of any psychotropic medication? It was unclear in the method. Although it is better to control the medication effect in patients without medication, it is unrealistic in clinical practice in treating BpD patients. Lack of standard medication treatment in severely depressed BpD patients may also against clinical treatment guideline. If the patients were also undergoing any psychotrophic agents, the author should list the medication.
5. Typos in page 6, line 232, “At the end of treatment”.
6. Lack of explanation of statistics in the methods. The author should explain the statistics in the method.
7. In the results, because of missing data in the repeat measurements, I suggested to use linear mixed model analysis and do not change statistic model in the same analysis.
8. The results shown in 3.3 and 3.4 section can be summarized in tables.
9. The subtitle in 3.4 section was wrong.
Author Response
We thank the reviewer for this fair and constructive evaluation. We agree with all limitations mentioned by the reviewer and also explicitly state those now in our revised manuscript. We further addressed the more specific comments as described in detail below
Please see the attachment.

Reviewer 2 Report
This study is considered valuable in that it provides evidence of rTMS. You did a great job. I will only point out some minor points.
1. First of all, references to the Research Ethics Committee should be included. The following sentence should be included, and even the review number of the Research Ethics Committee should be indicated.
For example) The study was approved by the institutional review board of ... (IRB No. ....).
2. As a limitation of the paper, it should be noted that all tools for evaluating a patient's depressive symptoms are self-reporting tools. Evaluation by an objective evaluation tool by a clinician such as Hamilton depression rating scales is omitted.
3. In bipolar disorder patients, comorbidity is important. In practice, comorbidity is quite common. Therefore, patients with comorbidities in this study need not be excluded from the study. However, it seems that the patients epidemiological characteristics including comorbidities should be included in Table 1.
Author Response
This study is considered valuable in that it provides evidence of rTMS. You did a great job. I will only point out some minor points.
We thank the reviewer for this positive and encouraging comment. We are happy to address the minor points as described in detail, below.
1. First of all, references to the Research Ethics Committee should be included. The following sentence should be included, and even the review number of the Research Ethics Committee should be indicated.
For example) The study was approved by the institutional review board of ... (IRB No. ....).
We thank the reviewer for pointing out the lack of clarity regarding the institutional review board. The data presented here were collected in a private clinical practice using treatment as usual (TAU) and no placebo control (open label). Therefore, IRB approval is not mandatory. Instead, we collected informed consent forms from all patients who agreed to participate and to have their (anonymised) data be used for scientific research. We clarified and stated this now explicitly in the methods section of the revised manuscript.
2. As a limitation of the paper, it should be noted that all tools for evaluating a patient's depressive symptoms are self-reporting tools. Evaluation by an objective evaluation tool by a clinician such as Hamilton depression rating scales is omitted.
This is an excellent point. We added this to our description of limitations in the revised version of our manuscript.
3. In bipolar disorder patients, comorbidity is important. In practice, comorbidity is quite common. Therefore, patients with comorbidities in this study need not be excluded from the study. However, it seems that the patients epidemiological characteristics including comorbidities should be included in Table 1.
This is a good point. We included this information in table 1 about personality disorder comorbidities.
Please see the attachment.
